# Visual Reconstruction of Ancient Coins Using Cycle-Consistent Generative Adversarial Networks

**Marios Zachariou \*, Neofytos Dimitriou \* and Ognjen Arandjelović \***

School of Computer Science, University of St Andrews, Scotland KY16 9AJ, UK

\*   Correspondence: marios.zachariou@hotmail.com (M.Z.); neofytosd@gmail.com (N.D.); ognjen.arandjelovic@gmail.com (O.A.)

**Abstract:** In this paper, our goal is to perform a *virtual restoration* of an ancient coin from its image. The present work is the first one to propose this problem, and it is motivated by two key promising applications. The first of these emerges from the recently recognised dependence of automatic image based coin type matching on the condition of the imaged coins; the algorithm introduced herein could be used as a pre-processing step, aimed at overcoming the aforementioned weakness. The second application concerns the utility both to professional and hobby numismatists of being able to visualise and study an ancient coin in a state closer to its original (minted) appearance. To address the conceptual problem at hand, we introduce a framework which comprises a deep learning based method using Generative Adversarial Networks, capable of learning the range of appearance variation of different semantic elements artistically depicted on coins, and a complementary algorithm used to collect, correctly label, and prepare for processing a large numbers of images (here 100,000) of ancient coins needed to facilitate the training of the aforementioned learning method. Empirical evaluation performed on a withheld subset of the data demonstrates extremely promising performance of the proposed methodology and shows that our algorithm correctly learns the spectra of appearance variation across different semantic elements, and despite the enormous variability present reconstructs the missing (damaged) detail while matching the surrounding semantic content and artistic style.

**Keywords:** deep learning; computer vision; Cycle-GAN; image reconstruction

## 1. Introduction

The aim of the work described in the present paper is to generate a realistic looking synthetic image of an ancient coin prior to suffering damage through wear and tear, from an image of an actual coin in a damaged state. This is a novel challenge in the realm of computer vision and machine learning based analysis of ancient coins, introduced herein for the first time. It is motivated by the value it adds in two key application domains. The first of these concerns hobby numismatists who would benefit from seeing what their coins looked like originally, as well as from being more readily able to identify them—a formidable task for non-experts. The second target audience concerns academic numismatists, who use information on ancient coins as well as contextual information of finds (geographic location of the find, manner in which the coins were deposited, which coins are found together, etc.). Being able to create synthetic images of less damaged coins could be of great benefit as a pre-processing step for automatic methods for coin analysis, such as identification [1] or semantic description [2], as the performance of such methods has been shown to be significantly affected by the state of preservation of the imaged coins [3].

## 2. Background: A Brief Introduction to Ancient Numismatics for the Non-Specialist

Learning about the distant past relies in large part on the use of artefacts in various forms such as tools, skeletons, pottery, drawings, scripts, and many others. The knowledge which emerges from these is inherently reliant on the process of interpretation—a challenging and interesting task both in practical and methodological terms. When it comes to the period widely known as classical antiquity, which is usually taken to have begun in the 8th century BC (foundation of Rome and the first Olympiad) and ended in the 6th century AD (ultimate fall of the Western Roman Empire and the decisive dominance of the Eastern Roman Empire), an artefact type of particular interest is that of a simple coin. Thus the term numismatics is used to describe both the formal, academic study of currency, as well as, colloquially, the popular hobby of collecting coins, tokens, and other means of payment.

There are many reasons why coins are important to historians. Firstly, by their very nature, they were widely used objects. Information on where certain coins were found provides valuable insight into migratory patterns of peoples, trading routes, cultural points of contact, etc. Moreover, coins themselves, by virtue of the content depicted on them, are inherently informative. Coins were used to mark important social events (e.g., the murder of Caius Iulius Casear, Roman conquest of Judea), architectural feats (e.g., Trajan's bridge, Colosseum), natural phenomena (e.g., comet sightings, Sun eclipses), as well as to spread political messages and propaganda (e.g., post-recession promises of better times ahead, emperor's divinity); see Figure 1. Indeed, some Roman emperors are only known from their coins, with no other historical record of their existence remaining (e.g., usurper Domitianus, following a coin find in Oxfordshire, UK; not to be confused with the 1st century emperor Domitian).

### 2.1. Basic Terminology

The specialist vocabulary of numismatics is extremely rich, and its comprehensive review is beyond the scope of the present article. Herein, we introduce a few basic concepts which are important for the understanding of our work.

Firstly, when referring to a 'coin', the reference is being made to a specific object, a physical artefact. It is important not to confuse it with the concept of a (coin) 'type' which is more abstract in nature. Two coins are of the same type if the semantic content of their obverses and reverses (heads and tails, in modern, colloquial English) is the same. For example, if the obverses are showing individuals (e.g., emperors), they have to be the same individuals, be shown from the same angle, have identical headwear (none, crown, wreath, etc.), be wearing the same clothing (drapery, cuirass, etc.), and so on. Examples are shown in Figure 2. Moreover, any inscriptions, usually running along the coin edge (referred to as 'legend'), also have to be identical, though not necessarily be identically arranged spatially letter by letter [4].

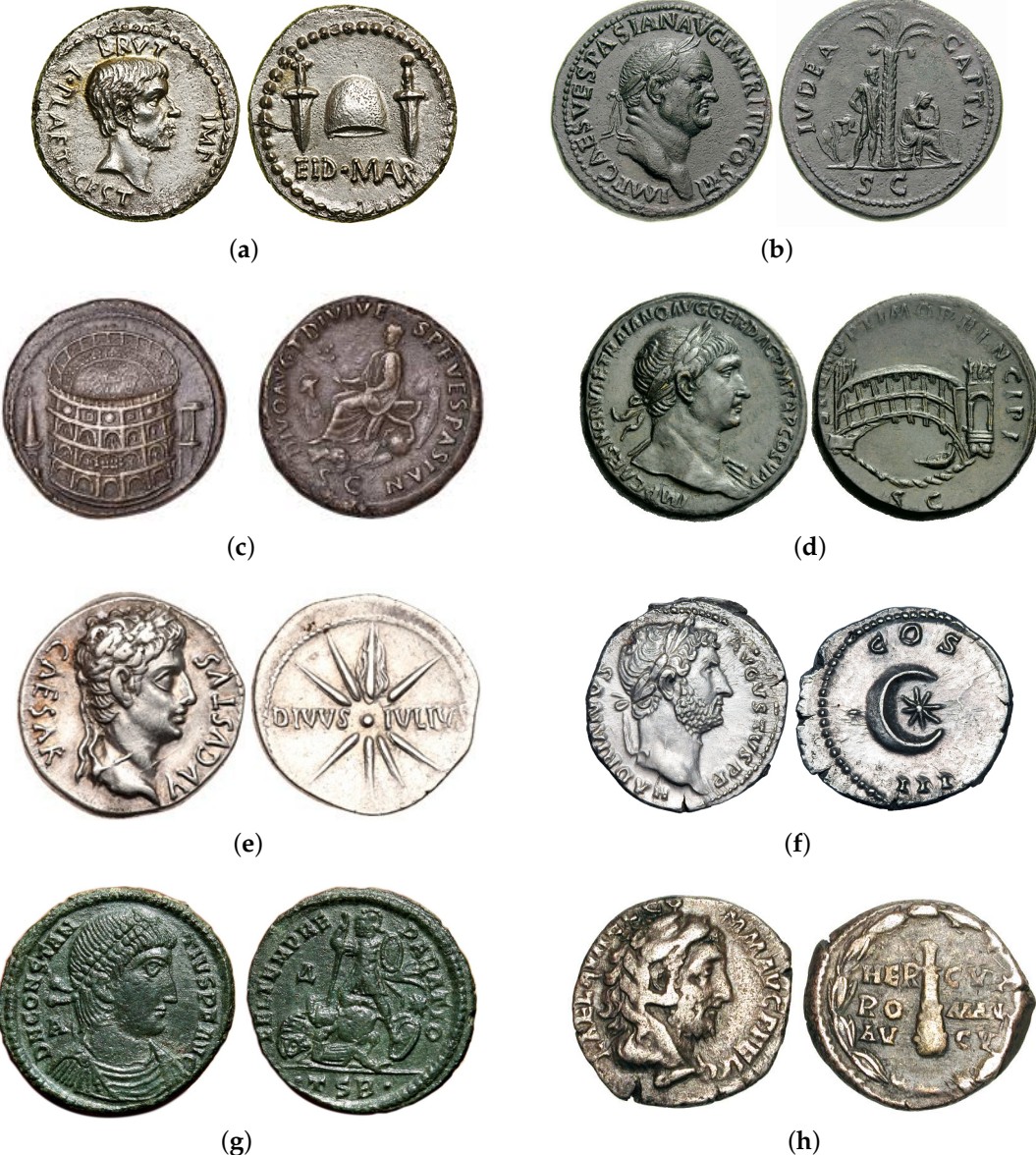

**Figure 1.** A small set of examples illustrating the breadth of messaging on ancient coins: (**a**) one of the most desirable Roman coins amongst collectors which celebrates the killing of Caius Iulius Caesar, (**b**) emperor Vespasian's coin commemorating the conquest of Judaea, (**c**) contemporary depiction of Colosseum, (**d**) Trajan's bridge over Danube, (**e**) the comet associated with the birth of Jesus by Christians, (**f**) solar eclipse observed during the reign of Hadrian, (**g**) the promise of better times (FEL TEMP REPARATIO) circulated across the empire weakened by the 3rd century economic crisis, and (**h**) Commodus shown as Hercules reincarnated.

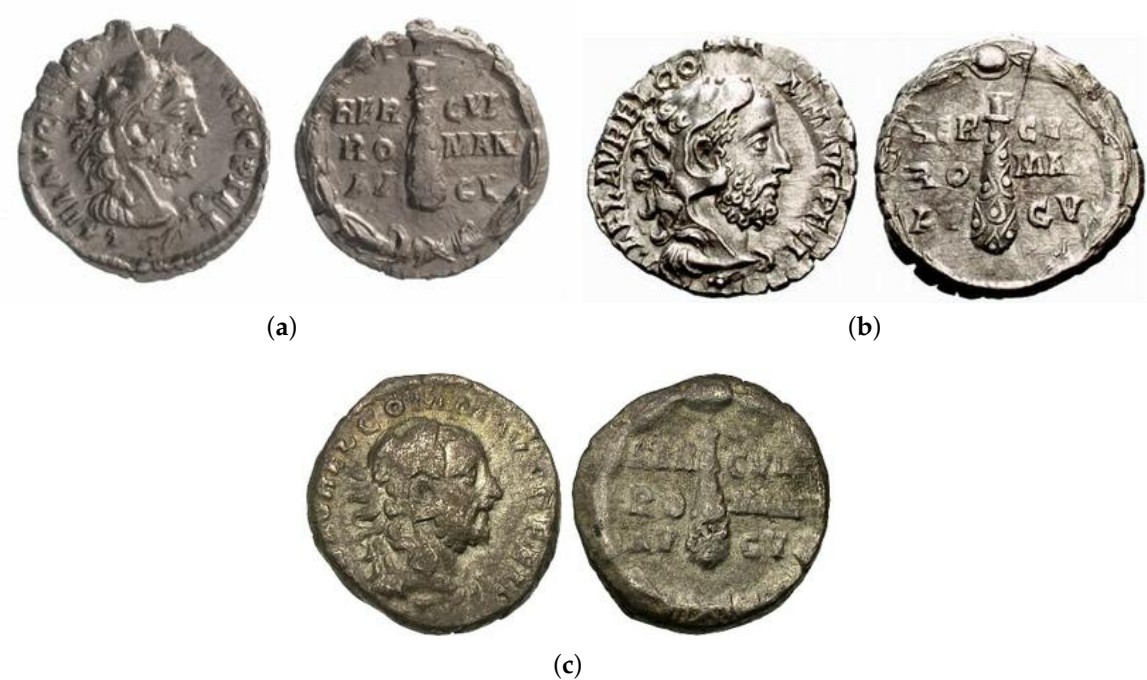

(a)                                    (b)

(c)

**Figure 2.** (**a**–**c**) Three specimens of the same coin type (silver denarius of Commodus, RIC 251). Observe that the exact placement of letters as well as artistic detail are different in all three cases, due to different dies used in the minting process. However, in terms of their semantic content, the motifs (legend, lion cladding, beard, club, etc.) on the obverses and reverses are identical.

*2.2. Grading*

An important consideration in numismatics regards the condition of a particular coin. As a millennium and a half to three millennia old objects, it is unsurprising that in virtually all cases they have suffered damage. This damage was effected by a variety of causes. First and foremost, most coins being used for day to day transactions, damage came though proverbial wear and tear. Damage was also effected by the environment in which coins were stored, hidden, or lost, before being found or excavated—for example, moisture or acidity of soil can have significant effects. Others were intentionally modified, for example for use in decorative jewellery. Figure 3 illustrates some of the common modes of damage.

The amount of damage to a coin is of major significance both to academic and hobby numismatists. To the former, the completeness of available information on rare coins is inherently valuable but equally, when damaged, the type of damage sustained by a coin can provide contextual information of the sort discussed earlier. For hobby numismatists, the significance of damage is twofold. Firstly, a better preserved coin is simply more aesthetically pleasing. Secondly, the price of the coin, and thus its affordability as well as investment potential, are greatly affected: the cost of the same type can vary 1–2 orders of magnitude.

To characterise the amount of damage to a coin due to wear and tear, as the most common type of damage, a quasi-objective grading system is widely used. Fair (Fr) condition describes a coin so worn that even the largest major elements are mostly destroyed, making even a broad categorisation of the coin difficult. Very Good (VG) coins miss most detail but major elements remain relatively clear in the outlines. Fine (F) condition coins show significant wear with many minor details worn through but major elements still clear at all the highest surfaces. Very Fine (VF) coins show wear to minor details, but clear major design elements. Finally, Extremely Fine (XF) coins show only minor wear to the finest detail.

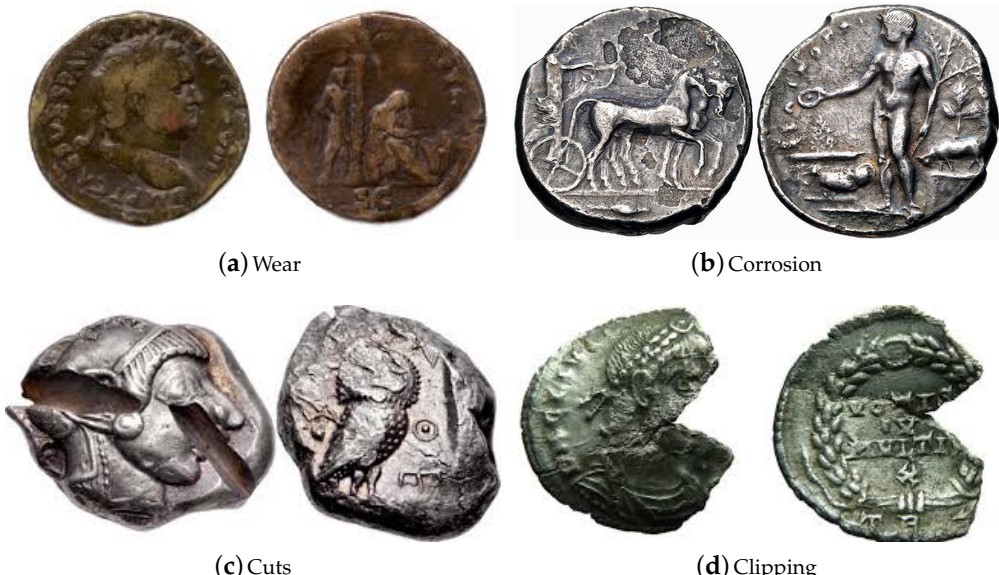

(**a**) Wear        (**b**) Corrosion

(**c**) Cuts        (**d**) Clipping

**Figure 3.** Examples of some of the most common types of damage suffered by ancient coins. (**a**) Worn out sestertius, a low denomination coin, which due to high weight and extensive use is rare in unworn condition. (**b**) Greek tetradrachma showing corrosion on the obverse. (**c**) Greek drachma with intentional contemporary cuts used to ascertain its authenticity and composition. (**d**) Clipped 4th century Roman siliqua, with major visual elements missing.

## 3. Related Work on Computer Vision and Machine Learning for Ancient Numismatics

Challenges emerging from the numismatics community first attracted the attention of computer vision and machine learning specialists some decade and a half ago, and since then, this interest has grown at an increasing pace, spurring an entire new sub-field of research. Most previous work has focused on one of the first tasks that one is confronted with when handing a new coin, namely the question 'what coin is this?'. More precisely put, given an image of a coin, the challenge is that of identifying the unique identifier of its type in a standard reference such as the magnum opus that is Roman Imperial Coins [5].

The early attempts at visual type matching of ancient coins reasonably attempted to apply existing object recognition approaches in the form of local features [6–9], which have shown good performance in a wide range of general object recognition tasks [10]. However, it quickly became apparent that their application in this domain is inadequate. The foremost fundamental problems which were identified stem from the similarity of local appearances across semantically unrelated elements of coins (unsurprisingly, given that coins are made of uniform material and exhibit little type specific albedo variability), and the loss of spatial information between local features [11]. Thus, local feature based approaches only showed some success in specimen matching and even that only in reasonably controlled conditions—those with little to no variability in illumination, and absent of clutter—invariance being needed solely with respect to in-image-plane rotations. An attempt to build upon local features and create compound, higher level features which combine local appearance and geometry achieved some (though limited) success [11], still performing far off requirements by virtually any practical application. Other approaches attempted to divide a coin into predefined regions, thus explicitly encoding some geometric information but the implicit assumption of perfect centering, registration, and flan shape was shown to lead to no performance improvement even on modestly sized data sets [3]. Although the legend can be a valuable source of information, relying on it for identification is problematic because it is often significantly affected by wear, illumination, and minting flaws [12] as well as major additional challenges which emerge in the

process of automatic data preparation, that is segmentation, normalisation of scale, orientation, and colour [13]. Recently, a number of deep learning approaches have been proposed [1]. The reported results, especially in the light of large data sets used, are highly promising and motivate further work in new approaches in this realm.

In practical terms, it is also important to note that nearly all existing work in this area is inherently limited by the reliance on visual matching [6–9,11] and the assumption that the unknown query coin is one of a limited number of gallery types against which it can be compared. However, this assumption is unrealistic due to the very large number of different coin types [5]—Online Coins of the Roman Empire, a joint project of the American Numismatic Society and the Institute for the Study of the Ancient World at New York University, lists 43,000 published types. Only few of these have been imaged. The effort, both in terms of labour and coordination (e.g., many types are extremely rare), to image them all is impractical. The first to make the point were Cooper and Arandjelović [2,14] who instead proposed an approach which uses deep learning and information from free form text descriptions of coins to learn the semantics of visual, artistically depicted elements. These are the first steps towards an automatic description of arbitrary, unseen specimens which may not have image exemplars in a gallery of known types.

## 4. Generative Adversarial Networks: A Brief Overview

A GAN in its simplest form ('vanilla GAN') comprises two CNNs, referred to as the Generator and Discriminator networks, which are trained jointly [15]. As the name suggests, given data in some input domain, the Generator synthesises data in the target domain [16]. On the other hand, the Discriminator tries to distinguish between real data in the target domain and synthetic data produced by the Generator [15,16]. The joint training of the two networks drives the Generator to improve in achieving realism of synthetically generated data and the Discriminator to become more nuanced in its discrimination. This process can be seen as a competition between two players, each trying to beat the other in a min-max game i.e., the two have adversarial objectives [17]. Some of the main advantages of employing GANs for this kind of cross-domain translation include the absence of need for Markov style sampling) [18,19] or a heuristic functions (e.g., pixel-wise mean square error) for representation learning [20].

### 4.1. Limitations of GANs

Notwithstanding the advantages that vanilla GAN-based approaches offer, as they are constituted now, they also exhibit some notable limitations. A detailed treatment of these is outside the scope of the present work, so herein, we discuss those that are of most relevance to our methodology.

Firstly, GANs have no means of driving the choice of modes that are generated from the data [18]. In an attempt to address this, a variation usually referred to as conditional GAN has been proposed and successfully applied in a wide range of tasks such as image from image generation [18,21], attribute driven image synthesis [22,23], captioning [24], and others [16,25–29]. Secondly, there is a danger of overfitting, for example by learning to map a set of images to any random permutation of the target output, producing a non-injective mapping [30].

### 4.2. Relevant Related Variations of GANs

As noted earlier, the present work is the first attempt to argue for the task of synthetic generation of coin images as a means of reconstructing the appearance of damaged ancient coins. Hence, no direct prior work in the field exists. However, for completeness we now identify and discuss conceptually related work which was done in other fields.

In terms of its technical underpinnings, the closest work in spirit to our method is the dual learning approach introduced by Xia et al. [31]. Their algorithm consisted of two GANs learning to translate natural language at the same time (side-by-side); in particular from English to French and vice versa. The key idea was that the model which accepts English as input can also evaluate the

output from the model that has English as its output (target domain), thereby creating a consistent boostrap of learning, reducing the amount of labelled data needed.

A major limitation of the algorithm of Xia et al. lies in the requirement that the two GANs are trained on monolingual data. In an attempt to overcome this limitation Zili et al. [21] introduced a method whereby the input domain is adaptive. Specifically, in their method the generative error is evaluated as the mean of absolute error between the input domain and the output domain. The success of the has since been demonstrated on tasks such as super-resolution [28], image generation from maps [16], and video based prediction [25].

An alternative approach was proposed in the form of the so-called Coupled GAN (CoGAN) [32]. Much like in Cycle-GANs [30] and Dual-GANs [21], CoGANs train two networks simultaneously without needing explicit pairing of input and target domain data. However, unlike the former two, in CoGANs the two networks are in no way connected i.e., they are completely independent in their learning. They instead learn the joint distribution of data from the two given domains. High level semantics in feature extraction can therefore be learnt by employing weight sharing. CoGAN can thus force the two networks to make sense of those semantics. The disadvantage is that weight-sharing techniques, although robust for CNNs [33] and similar approaches to solutions [34,35], tend to lock the network to domain specific as weight sharing has to be set a priori.

## 5. Technical Contribution: The Proposed Method

Our cycle-GAN two generators and two discriminators in a cyclic manner: each generator learns the mapping from one domain to the other, namely from F to XF, and XF to F respectively. In other words, the first generator is learning to produce synthetic images of Extremely Fine coins using as input images of Fine coins. The second generator does the same but in the opposite direction, in effect learning to model the degradation process experienced by ancient coins over time. This is illustrated in Figure 4.

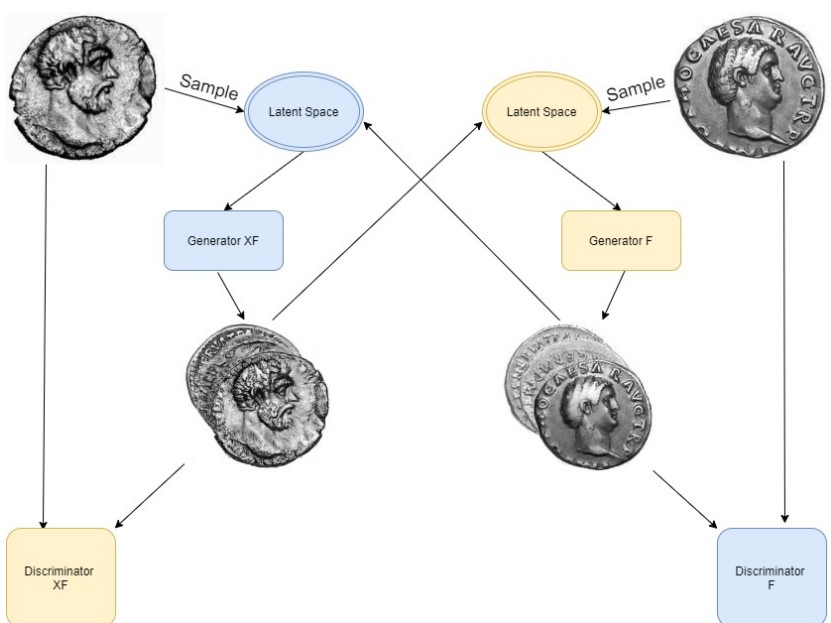

**Figure 4.** Conceptual diagram of the key components and processing stages of our network.

### 5.1. Network Architecture

We adopt the use of two ResNets, one with 6 residual networks having as input $128 \times 128$ pixel images, and one with 9 residual layers and $256 \times 256$ pixel input images. Although Batch Normalisation

is used in transitional layers by most work in the literature, we found that on the present task Instance Normalisation was superior and was hence adopted instead. An additional advantage of following this approach is to be found in the absence of need for large batches required by Batch Normalisation [15]. For the Discriminator we used an architecture which is much shallower than that of the Generator. Our Discriminator model is made up of five layers inspired by the PatchGAN [27,30]. We used $29 \times 29$ pixel output.

### 5.2. Training and Parameter Choice

We found that in comparison with most work in the literature, to ensure successful training a higher number of learning epochs was required in the present task. For example, we found that 200 epochs used by Zhu et al. [30] were insufficient; instead we used significantly more, 50%, i.e., 300 epochs. This is likely a consequence of the highly challenging nature of our data and the ultimate task, that is the subtle nature of the artistic style and semantic elements depicted on Ancient Roman coins.

### 5.3. Gradient Optimiser and Batch Size Choice

Considering the novelty of our application domain and thus the lack of relevant information from prior research which could inform our optimiser choice, we considered it important not to follow automatically the consensus and instead compare different optimisers' performance: Stochastic Gradient Descent (SGD) [36] and Adaptive Moment Estimation (Adam) [2]. The momentum parameter for SGD (specifically, the variant termed Online SGD [36]) was set to 0.9, and the Adam parameters $\beta_1$ and $\beta_2$ to 0.5 and 0.99 respectively. Figure 5 illustrates our typical findings when the two approaches are compared with one another, whereas Figure 6 summarises the effects of batch size on the performance of SGD. In terms of image reconstruction diversity, the use of Adam was found to match or outperform SGD. This is consistent with the observation that the Generator losses with SGD decayed significantly more slowly. Consequently, the adversarial loss also ceases to change, which suggests that at that stage little new information is being passed to the Generator. In conclusion, our findings clearly motivated the use of Adam as the preferred optimiser choice, which unless stated otherwise should be assumed for all results reported hereafter.

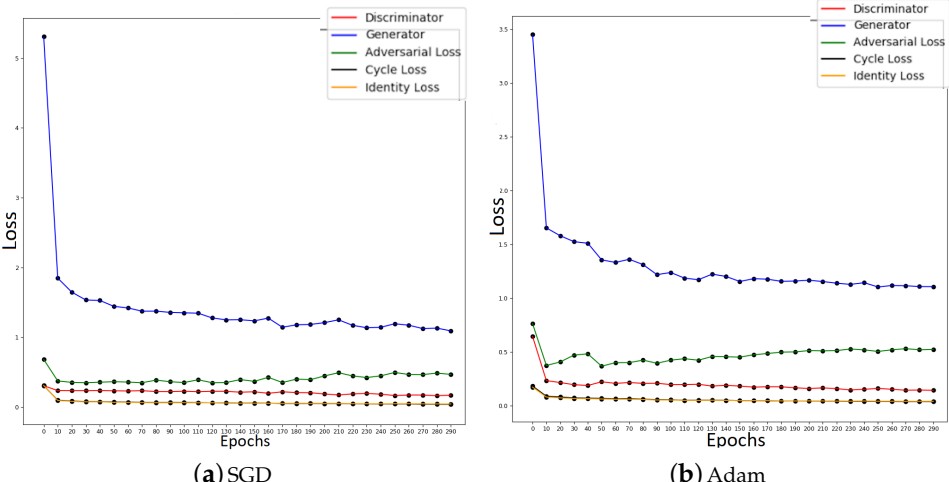

**(a)** SGD      **(b)** Adam

**Figure 5.** Comparison of loss variability during training using a Stochastic Gradient Descent and Adam. All parameters were set to be equal for both optimisers.

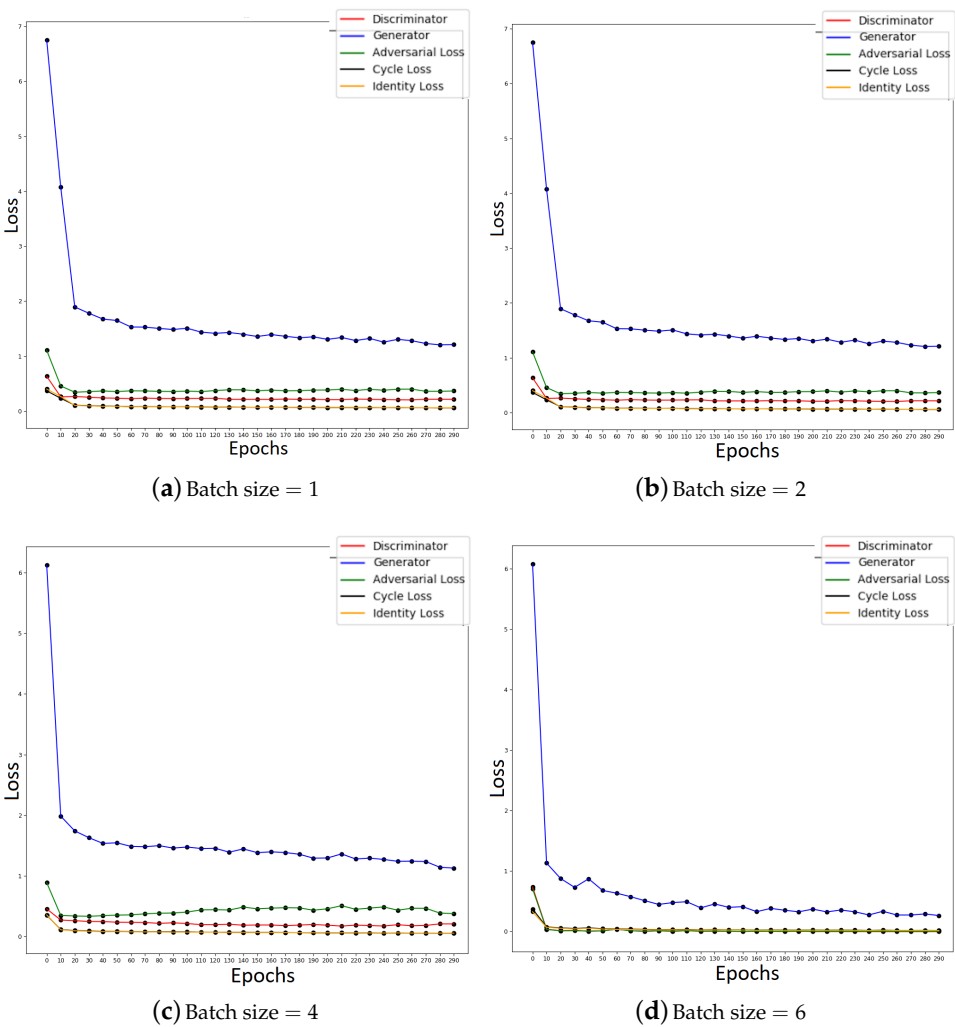

**Figure 6.** The effect on loss of increasing batch size, using the Stochastic Gradient Descent optimiser.

## 5.4. Learning Rate

Unlike most previous work, which sets the learning rates of the Discriminator and the Generator to be equal (usually $2 \times 10^{-4}$), in this work, we deviate from this tradition substantially. In particular, we found that on the problem under consideration, with equal learning rates the Discriminator was prone to overfitting, consistently learning around the same mode in the data. We observed that this often corresponded to the pixel intensity contrast which makes sense: with coins being metallic and exhibiting specular behaviour, the level of preserved detail greatly affects the manner in which light is reflected off their surface. This motivated a substantially higher Generator learning rate for the Discriminator. What is more, since the learning gradient is more accurately computed on large batch sizes, a higher learning rate makes sense in that larger adjustment (steps) can be more confidently made [36–38]. In the end, we set the learning rate of our Discriminators to be $2 \times 10^{-4}$ and that of our Generators $6 \times 10^{-4}$.

## 5.5. Data Augmentation and Normalisation

Being created for use in auctions by reputable auction houses, unlike those sourced from user driven web sites like eBay [13], the images in our data set are reasonably homogeneous as regards geometry and photometry. Nevertheless, there remain a number of extrinsic sources of variability

which do not cause problems when the images are used by humans who partake in auctions, but which can affect machine learning significantly.

Firstly, as regards geometric variability, the rotational alignment (registration) of coin motifs is not perfect. In part, this is a consequence of different die designs and thus the fact that the correct orientation is fundamentally ill-defined, and in part by the lack of practical need (in terms of what the images were acquired for) for absolute precision. Similarly, there is variation in scale which again is inconsequential in the context of human observation but salient when the images are to be used within an automatic framework. Non-uniformity in scale can also be due to fundamental factors, such as inconsistent flan size effected by the primitive minting process, and the aforementioned practical reasons. Similar observations can be made with respect to confounding photometric heterogeneity, the illumination conditions between lots can vary, especially as images were acquired at different times, by different photographers, in different studios, and using different imaging equipment [39].

To facilitate robust learning in the presence of the described confounding factors, we perform data augmentation prior to training our networks. This is a common strategy which has proven successful both in deep learning [1] and machine learning in general [40,41].

Specifically, herein, we augment our original data set with synthetic images simply generated from a random subset of the original images. In particular, we apply random rotations and rescaling by drawing the relevant parameters, namely the rotation angle and the scaling factor, from uniform distributions over the ranges of respectively $\pm 15^\circ$ and 0.7–1.4. Photometric variability is dealt with not by data augmentation but rather normalisation, whereby all data is photometrically rescaled to the same mean and unit variance.

## 6. Data, Preparation Thereof, and Automatic Label Extraction

In the previous section, we focused on the main technical underpinnings of the approach proposed in the present paper. The data, that is, images, used for training, were implicitly assumed to be in a canonical format and correctly labelled in terms of class membership (coins in either Fine or Extremely Fine condition). However, meeting these requirements in the real world is non-trivial and in fact rather challenging. Hence, in this section, we describe another important contribution of this work, in the form of a preprocessing pipeline which starts from a readily available, large data set of images and unstructured meta data collected "in the wild", and produces data which is correctly labelled and normalised in a manner which meets the requirements of the framework proposed in the previous section.

### 6.1. Raw Data

Our unprocessed data set costs of 100,000 lots of ancient coins sold by reputable auction houses across the world (mainly based on Europe and the USA). These were kindly provided to us for research purposes by the Ancient Coin Search auction aggregator https://www.acsearch.info/. Each lot comprises an image of a coin (the images of the obverse and the reverse of the coin come concatenated, forming a single image) and the associated free form text description of the coin.

#### 6.1.1. Image Preparation

As noted previously, the raw images in our data set contain images of coin obverses and reverses concatenated, just like those in Figure 1. This makes the extraction of obverses, which we focus on in this work, rather straightforward. It is likely that even a simple vertical splitting of raw images halfway would suffice but considering that we could not manually inspect the entire data set due to its size, we opted for a more robust approach. In particular, we first detect the vertical band separating the obverse from the reverse by finding the region of extrema in the vertical integral (i.e., along the *y*-axis) of pixel intensities. The centre point of this band is set to split the raw image, with the left hand part (by the universally adopted convention) corresponding to the obverse.

It should also be noted that although the original images are in colour, we treat them as being greyscale for two reasons. The foremost one is to be found in the conceptual novelty of the present work and thus the desire to start with a simpler challenge first, so as to understand how tangible the task is and the potential unforeseen difficulties. The second one is of a computational nature—learning in greyscale is faster and necessitates less data, and again given the pioneering nature of the work, it was not clear a priori how many data are needed to tackle the challenge adequately. This consideration also motivated conservatism in approach.

### 6.1.2. Label Extraction

Each of the sale lots in our database contains a free form text description of the lot. These descriptions vary in structure, length, and content, and can be in different languages. English language descriptions are most common but there is a substantial number of entries in German and French, and a smaller number in Spanish and Italian. To illustrate some of these challenges, we show a few different descriptions in Figures 7 and 8.

Recalling that our goal is to produce synthetic images of coins prior to their being damaged, the labels that we wish to extract from this free form text are the condition descriptors of coins. In particular, for reasons which will be explained in the next section, we are looking for coins in Fine and Extremely Fine conditions.

An additional difficulty in extracting this information from the given textual descriptions emerges from the lack of uniformity in how the condition of a coin is stated, i.e., whether it is in its full or abbreviated form; see Table 1.

The first part of our label extraction process uses text only. In particular we use a regular expression that matches the patterns of the target condition descriptors in different languages, as per Table 1. In most cases this step alone is sufficient but there remain a number of ambiguous cases. For example, in some instances breaks in a coin's legend can produce fragments which match a specific abbreviated descriptor. Say, to denote a spatial break in a legend such as 'FELICITAS', the legend may be written as 'F ELICITAS', the leading 'F' thus being incorrectly identified as corresponding to a grade descriptor. Hence, we follow up on this first, purely text based step, by an additional one which uses complementary information: the actual image of a coin.

**Table 1.** Standard abbreviations in different languages for the two coin grades referred to as 'Fine' and 'Extremely Fine' in English. The languages shown account for nearly all of our data, reflecting the locations of the most popular auction houses and their target audiences. For clarity, following each abbreviation is the corresponding expression in full (in brackets).

| | | |
|---|---|---|
| US English | F (Fine) | EF (Extremely Fine) |
| UK English | F (Fine) | XF (Extremely Fine) |
| French | TB (Trés Beau) | SUP (Superbe) |
| Spanish | BC+ (Bien Conservada+) | EBC (Extraordinariamente Bien Conservada) |
| German | S (Schön) | VZ (Vorzüglich) |
| Italian | MB (Molto Bello) | SPL (Splendido) |

```
Hadrian
Hadrian, 117 - 138 n. Chr. Denar 134 - 138 n. Chr. Rom. 3.01 g. Vs.: HADRIANVS AVG
COS III P P, Kopf n. r. Rs.: FIDES PVBLICA, Fides mit Obstkorb u. Ähren. RIC 241a;
C. 716; BMC 627; Strack 237. Etwas rau, vz/ss
```

```
HADRIEN(11/08/117-10/07/138)Publius lius Hadrianus Quinaire N v21_2511
Date : 119 ou 121
Nom de l'atelier : Rome
Métal : argent
Diamètre : 14mm
Axe des coins : 6h.
Poids : 1,56g.
Degré de rareté : R2
Etat de conservation : TB+ Prix de départ : 150    Estimation : 200
Prix réalisé : 150
Commentaires sur l'état de conservation : Exemplaire parfaitement identifiable et
lisible en bon état pour un type monétaire souvent défecteux. Beau portrait. Flan
éclaté à 12 heures au droit. Une épaisse patine grise recouvre l'ensemble de la
monnaie. N dans les ouvrages de référence : C.1137 (10f.) - RIC.108  - BMC/RE.23
5  - H.1/133 ou 196 - RSC.1137
Pedigree : Cet exemplaire provient d'une vieille collection du Sud de la France.
Titulature avers : IMP CAESAR TRAIAN - HADRIANVS AVG.
Description avers : Buste lauré d'Hadrien à droite, drapé sur l'épaule gauche (O*2).
Traduction avers : 'Imp Csar Traianus Hadrianus Augustus', (L'empereur césar Trajan
Hadrien auguste).
Titulature revers : P M T-R P - COS III.
Description revers : Victoria (la Victoire) assise à gauche, les ailes déployées,
tenant une couronne de la main droite et une palme de la main gauche.
Traduction revers : 'Pontifex Maximus Tribunicia Potestate Consul tertium', (Grand
pontife revêtu de la puissance tribunitienne consul pour la troisième fois).
Commentaires : Ce quinaire est frappé soit à l'occasion de la deuxième distribution
d'argent effectuée par Hadrien en 119, liée au commencement de la troisième année de
règne et à l'annulation des dettes avant la consécration de Matidie soit à
l'occasion de la troisième libéralité en 121 accompagnant le ''Natalis Urbis''
(anniversaire de la naissance de Rome 21 avril 753 avant J.-C.).
Historique : Hadrien est né en 76 à Italica. Pupille de Trajan, il épousa en 100 la
petite-nièce de l'empereur, Sabine, et fit carrière dans l'état-major de l'empereur,
en particulier lors de la campagne dacique. En 117, il succéda à Trajan, et voyagea
pendant vingt ans, visitant l'ensemble de l'Empire, le seul empereur à l'avoir fait.
En 122, Hadrien se rendit en Espagne. Sans enfant, il choisit d'abord Aélius pour
lui succéder en 136, mais ce dernier mourut le 1er janvier 138. Hadrien adopta alors
Antonin le 25 février et choisit lui-même Marc Aurèle et Lucius Vérus comme
héritiers d'Antonin. Il décéda le 10 juillet 138.
```

**Figure 7.** Typical free form text descriptions of lots (coins) in our data set. Also see Figure 8.

```
Roman Imperial Coins
Hadrian. A.D. 117-138. AR denarius (18 mm, 3.55 g, 6 h). Anomalous ('Eastern')
issue, A.D. 128/9. HADRIANVS AVGVSTVS P P, laureate head of Hadrian right / C-OS
III, Aequitas standing facing, head left, holding scales and cornucopiae. Cf. RIC
339 (Rome); BMC 1035; RSC 382. Lightly toned. Nice very fine. Ex CNG E262 (17
August 2011), 334. The 'Eastern' mint denarii of Hadrian are all quite rare. There
are two large assemblages of types published, but unfortunately neither are
comprehensive: BMCRE vol. II, pp. 372-81, ppl. 68-71, and the section on Hadrian's
Imperial coinage at the Beast Coins website
(http://www.beastcoins.com/RomanImperial/II/Hadrian/Hadrian.htm). With the
exception of the small subset of Antioch mint issues, Metcalf prefers to call
these non-Rome mint denarii of Hadrian ""anomalous"" as they do not follow the
same agenda as Hadrian's cistophori (Hadrian's cistophori combine Roman legends on
the obverse with local Greek types on the reverse, the purpose being ""to restore
federal government in the provinces and use the coinage as a tradition of the past
functioning in a contemporary context"" (Beastcoins.com). In addition to BMCRE and
the Beast Coins website, anyone wishing to study these anomalous denarii further
should consult both P. L. Strack, Untersuchungen zur römischen Reichsprägung des
zweiten Jahrhunderts (3 vols. Stuttgart, 1931-1937), and the Michael Kelly
Collection of Roman Silver Coins (Spink, 18 November 1997), lots 1033-1042.
```

**Figure 8.** Typical free form text description of lots (coins) in our data set. Also see Figure 7.

Intuitively, our motivation stems from the understanding of the process which causes damage to ancient coins. Specifically, as summarised in Section 2.2 the type of damage relevant here (and indeed the most commonly observed one) is mainly caused by 'wear and tear' (coins rubbing against one another while carried in moneybags, exchanged on hard surfaces) and environmental abrasion, both first and foremost affecting fine detail. In the context of the corresponding digital images, this translates to a loss of high frequency detail—for a long time a widely exploited observation in a variety of image processing tasks. This readily translates into a simple algorithm whereby discrimination between higher and lower grade coins—in our case those in Extremely Fine and Fine condition respectively—is performed by thresholding on the image signal energy content above a certain frequency (we used frequencies higher than one sixth of the highest possible frequency). We found that this complementary approach works extremely well in resolving labelling ambiguities when they occur in the first, text based step.

## 7. Evaluation

In many recent applications of GANs for cross-domain translation, the evaluation process is challenging. The key reason for this lies in the practical (if not inherent) impossibility of obtaining the ground truth against which synthetically generated data can be compared. This methodological challenge is particularly apparent in the context of the problem at hand: ancient coins imaged today were produced millennia ago and each coin being different from all others (even if struck with the same die, due to the primitive minting process used at the time), it is impossible to know what exactly it looked like before being damaged. Hence, the primary way of evaluating the result has to be subjective and rely on expertise of an experienced numismatist who is able to assess the realism of the artistic style of the coins in synthetic images, identify the accuracy of fine features of recovered (or indeed non-recovered) semantic elements, etc. To this end our results were inspected by a numismatist with nearly 30 years of experience in the study of Ancient Roman coins in particular.

A sample of typical results which illustrate the performance of our method is shown in Figure 9. Each example shows the input image of a real, damaged coin in Fine condition on the left, and the corresponding synthetically generated image of the same specimen with lesser damage on the right. It is readily apparent even to a non-numismatist that our results are extremely promising. Note how a wide range of fine details, entirely absent in the original images are reconstructed, from strands

of hair (correctly matching the depicted person's hair style, e.g., curls, strands, or waves) and facial features, to details on bodily garments (draperies and the like) and head ornaments (wreaths in this case). For emphasis, a magnification of a particularly interesting region of the image in Figure 9d is shown magnified in Figure 10. The region shown corresponds to one of the parts of the coin (a silver denarius of emperor Trajan) which contains some of the highest surface points. These are most prone to wear and tear damage, especially when they feature fine detail. The damage is severe in the example shown, as evident from the region in the input image; see Figure 10a. However, the corresponding reconstruction in Figure 10b contains remarkably restored detail. Moreover, the reconstructed detail is stylistically impressively accurate. It is clear, first and foremost, that our algorithm has correctly learnt the spectrum of appearance variation of hair specifically—the reconstructed detail *is* hair, rather than some arbitrary detail which may be found elsewhere on a coin. What is more, the style of hair is correct, despite the enormous variability across different individuals depicted on common obverses (locks, braids, waves, curls, etc.). Lastly, the missing detail fits the surrounding context well. All of this is especially impressive considering the relatively low resolution (in the context of the frequency of fine coin detail we are interested in) of the images, as readily apparent in Figure 10. As demonstrated by previous work in the field [3], the loss of this kind of detail can have a profound effect on the performance of computer vision based ancient coin recognition algorithms, providing evidence for one of the key premises motivating the present contribution—that of using the proposed method as a pre-processing step to further automatic analysis.

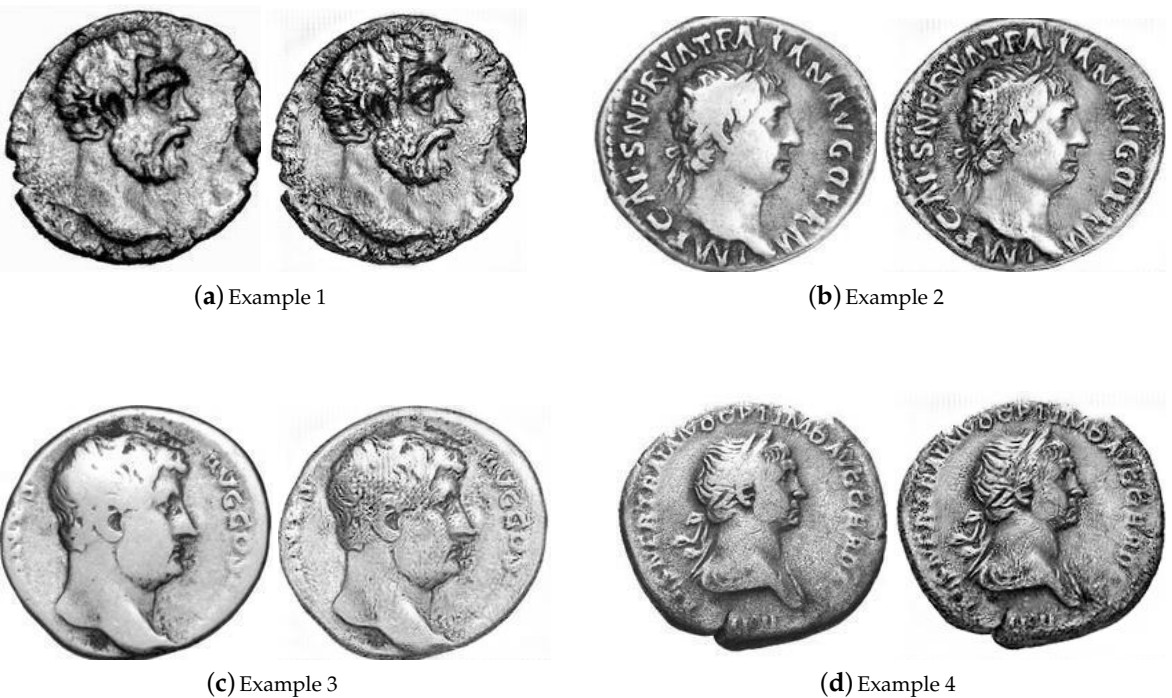

(**a**) Example 1

(**b**) Example 2

(**c**) Example 3

(**d**) Example 4

**Figure 9.** *Cont.*

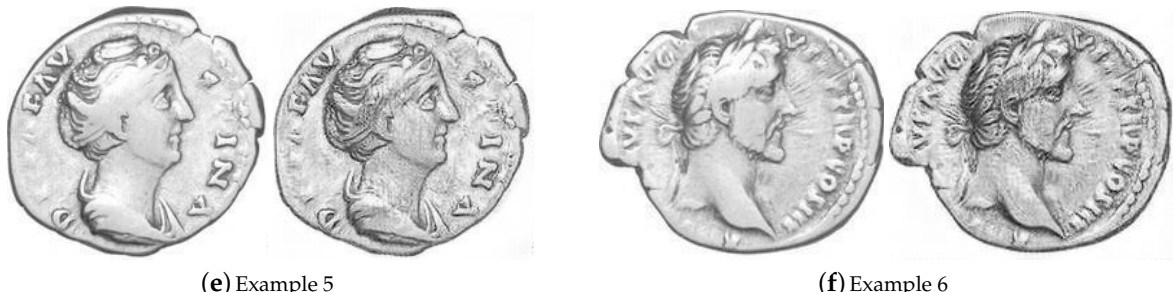

(**e**) Example 5　　　　　　　　　　　　　　　　　　　　(**f**) Example 6

**Figure 9.** Typical results obtained by applying our method on images of six different lower grade (Fine) ancient coins. Each subfigure comprises the original, input image of a real, damaged coin on the left, and the corresponding synthetically generated image of the same specimen with lesser damage on the right. Impressive reconstruction of appearance is clear, showing that our algorithm is correctly learning the range of appearances that different coin elements exhibit (compare with the work of Cooper and Arandjelović [2,14]), and is appropriately applying this model to match the overall visual context. Also see Figure 10.

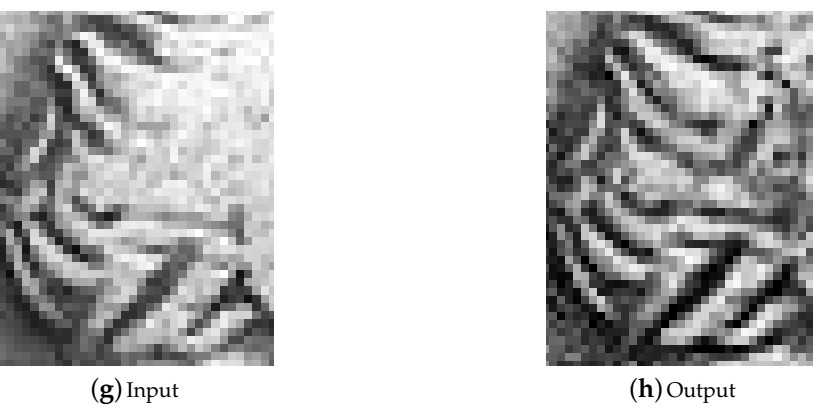

(**g**) Input　　　　　　　　　　　　　　　　　　　　　　(**h**) Output

**Figure 10.** Magnified detail from the example shown in Figure 9, featuring a denarius of emperor Trajan. It is clear that our algorithm has correctly learnt the spectrum of appearance variation of hair and appropriately reconstructed the detail missing due to damage of the coin in the input image, matching the style of hair and the surrounding context.

## 8. Conclusions and Future Work

In this paper, we introduced a new problem to the field of computer vision and machine learning applications in ancient numismatics. Thus, the first contribution of the work lies in the conceptual argument for the need for this problem to be addressed. In particular, our goal was to perform virtual reconstruction of an ancient coin, that is, given an image of a damaged coin to synthesise an image of the coin more closely resembling the specimen in its original (minted) state. We motivated this by recognising the potential for use of this synthetic imagery, for example as input to automatic coin type matching algorithms, or for study and visualisation purposes. Our second contribution is a technical one and it consists of a deep learning based algorithm designed as the first attempt to solve the problem. In particular, we introduce an algorithm which uses adversarial learning, with a Generative Adversarial Network at its heart, as a means of learning to model phenomenologically the processes which effect damage on ancient coins. To facilitate this learning, we also describe a methodology for obtaining correctly labelled data from unstructured raw data 'in the wild', needing no manual intervention. Our analysis of the method's learning performance also makes a series of contributions which are of benefit to researchers in the field considering the novelty of the task at hand—we found that the unique features of the type of data of interest required rather different design choices from those made in seemingly related applications in other domains. Lastly, we demonstrated the effectiveness of the proposed algorithms empirically. Impressive results were obtained, evident from the correct

learning of the spectra of appearance variation of different semantic elements on coins. Despite the enormous variability present, missing detail was successfully reconstructed in a manner which respects the artistic style of ancient Roman coins and the broader surrounding semantic content.

In addition to the specific, direct contributions summarized above, the importance of the present work lies in its opening new avenues for further work. We highlight a few which are on the top of our list but expect that the wider community will come up with an even broader range of ideas. Firstly, we would like to analyse the performance of our method on Ancient Greek and Celtic coins. In the present paper we chose to focus on Roman Imperial coins for several reasons. Firstly, thus far they have been the focus of most computer vision and machine learning based work in ancient numismatics. Abundant training data is also more readily available for this category of coins (they are more numerous and are more widely collected by hobby numismatists). Secondly, notwithstanding the artistic variability emphasised earlier, it can be reasonably expected that there is a greater degree of style standardization across mints of a single empire, like Rome, than across different different city states which comprised what is colloquially understood as Ancient Greece (let alone across different Celtic tribes, see Figure 11). Understanding the performance of the proposed method when applied to Ancient Greek and Celtic coins would likely bring insight as to how the method would be improved further. As regards future work pertaining to the technical underpinning of our method, our immediate plan is to investigate different forms of normalization. Specifically, while in this work we employed Instance Normalization, recent reports in the literature suggests that benefits can be conferred by improved Batch Normalization on small batches or the use of different types of normalization such as Group Normalization or Ghost Batch Normalization.

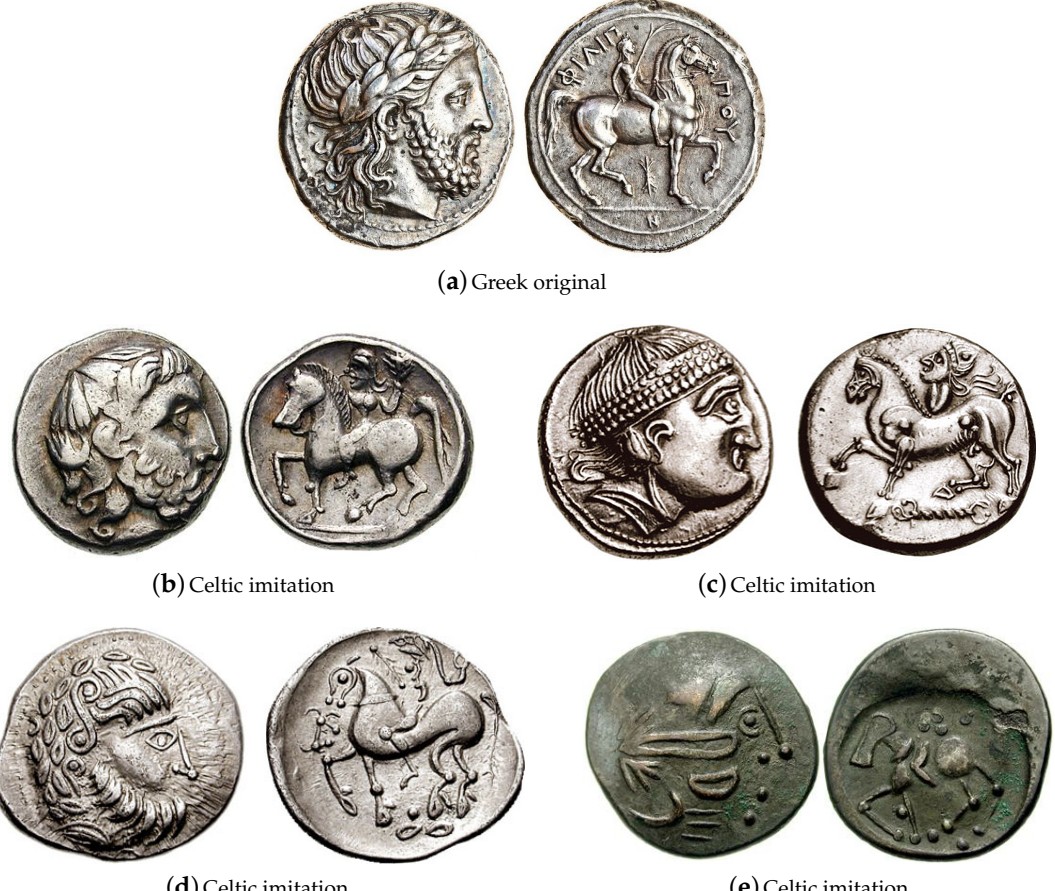

(**a**) Greek original

(**b**) Celtic imitation       (**c**) Celtic imitation

(**d**) Celtic imitation       (**e**) Celtic imitation

**Figure 11.** (**a**) Original Greek Macedon tetradrachm of Philip II (342–336 BC) and (**b**–**e**) different Celtic imitations (circa 3rd–2nd Century BC) thereof.

**Author Contributions:** All authors have contributed to all aspects of the described work. All authors have read and agreed to the published version of the manuscript.

**Funding:** This research received no external funding.

**Acknowledgments:** The authors would like to express their gratitude to Simon Wieland of acsearch AG (www.acsearch.info.) for kindly providing us with the data used in this research.

**Conflicts of Interest:** The authors declare no conflict of interest.

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
