# Peer review of "Visual Reconstruction of Ancient Coins Using Cycle-Consistent Generative Adversarial Networks"

_sci, doi:10.3390/sci2030052_

Round 1

Reviewer 1 Report

This paper was both well-written and well thought out. Certainly an excellent example of the application of GANs with a successful result. I frankly have no negative or constructive comments for you as the paper could readily be used as a tutorial for GANs application!

Author Response

This paper was both well-written and well thought out. Certainly an excellent example of the application of GANs with a successful result. I frankly have no negative or constructive comments for you as the paper could readily be used as a tutorial for GANs application! We are most grateful to the reviewer for the kind words - we truly appreciate them! Thank you for your time and effort.

Reviewer 2 Report

The paper was also submitted to a major European Computer Vision Conference in 2020, which is a violation against the MDPI standards (and might be seen as unethical behaviour):

Manuscripts submitted to MDPI journals should meet the highest standards of publication ethics:

  • Manuscripts should only report results that have not been submitted or published before, even in part.
  • Manuscripts must be original and should not reuse text from another source without appropriate citation.

Author Response

The paper was also submitted to a major European Computer Vision Conference in 2020, which is a violation against the MDPI standards We can assure the reviewer that there was no violation of policy at any stage, though we can see why the reviewer could have thought otherwise (as a side note, to avoid confusion it would have been more collegial to approach the authors directly to ascertain facts before making accusations). In particular we believe that the reviewer may not have understood that this is a post-acceptance review, and not a pre-acceptance one - the process is confusing to us too, no doubt.

Reviewer 3 Report

This submission is a very interesting paper with a very applied topic and falls in the scope of the journal. Although, it has some potential for improvement.

Introduction

  • The general description of the problem (Introduction) and the description of its importance for the science and the society could be further improved.
  • The degree of innovativeness of the methodological approach is not convincingly demonstrated. Some more details about its innovative features could further improve the quality of this paper.
  • Why is this paper likely to be cited in the future?

Method

  • A bit more text regarding the originality of this work and why it contains new results that significantly advance the research field.

Results

  • I believe that adding a bit more text on why the results of the method are satisfactory (evaluation approach) will increase the quality of this work
  • Could the results be more satisfactory if you have changed something in the methodology?
  • Are the results (or the method) sensitive to this specific study area?

Discussion

  • In the Discussion section I would have wished to see more information on the actual meaning of the findings and how the results add to the broader topic as well as the specific scientific field

Conclusion

  • The "Conclusions" section, could be further improved by describing the importance of this work, the highlight of potential further development of this methodology.

Author Response

This submission is a very interesting paper with a very applied topic and falls in the scope of the journal. We are genuinely very thankful to the reviewer for the positive comments.  Thank you both for these and the constructive feedback which we did our best to address to your satisfaction.  We trust you will find our revision even better, following its improvements effected by revisions in accordance with your feedback. Why is this paper likely to be cited in the future? We do that this is explained and emphasised in the manuscript. In particular, as stated therein (already in the abstract and then reiterated in more detail in the main text) this is the first work to recognize the very possibility of performing virtual reconstruction of ancient coins and the first work to recognize the practical potential in attempting to do this (visualization, pre-processing input to classification methods, etc.), and then naturally the first one to propose a solution to it and demonstrate its promise. Could the results be more satisfactory if you have changed something in the methodology? In the text of our manuscript we discuss systematically various design choices, stressing when and (crucially) why these differ from those in the existing literature employing technically related methodologies. We are sure that there always is opportunity for improvement and we certainly hope that our contribution will attract more research in the area, which will no doubt improve on our results.  That being said, if we actually already knew how to do so at this stage, we would have done and reported it. Following the reviewer's other comments, the revision now has more suggestions for future work and possible future improvements, which is perhaps what the reviewer meant here too. Are the results (or the method) sensitive to this specific study area We are not exactly sure what the reviewer is referring to as "this specific study area".  Is the reviewer referring to "ancient numismatics" or "Roman Imperial coins"?  In both cases, and indeed universally so when it comes to the learning of the type adopted in our paper, the answer is yes, as the ability to learn a specific characteristic of data depends on the breath of variation across data, its nature, the practicability of obtaining sufficient training data, etc.  The purposeful domain specific design choices have been highlighted at several stages (e.g. in Section 5.4) with explanations giving the reader further insight as to the phenomena which explain the differential design.  But we do agree that this is worth adding a further note on in the penultimate section and have therefore amended the manuscript accordingly. In the Discussion section I would have wished to see more information on the actual meaning of the findings We genuinely apologize but we again do not quite understand what the reviewer means by "the actual meaning of the findings".  As we noted when we motivated the work and hence commented upon in the discussion, we aim to create synthetic reconstructions of coins.  The "meaning" is the recovery of missing detail, again as discussed in the paper, and the importance of this is stated already in the manuscript: one can see what the coin looked like before damage took place (visualization) which is important for educational purposes and general appreciation of coins (e.g. by collectors), or indeed use this higher quality synthetic imagery as input to automatic classification and analysis algorithms (again, as stated in the manuscript, including comments on the importance of coin condition in this context with suitable references evidencing this). Nevertheless, to emphasise this further the section has been amdended. The "Conclusions" section, could be further improved by describing the importance of this work, the highlight of potential further development of this methodology. We are genuinely grateful for this suggestion as it made us sit back and reflect further on our work, and come up with actual new ideas! The revision has been amended accordingly - thank you again!

Reviewer 4 Report

I guess as the majority of reviewers have approved the manuscript from the basis of scientific quality we should therefore be in a position to accept it

Round 2

Reviewer 3 Report

it is ok now. I believe that no further changes are requested.